# Auto-Scaling Vision Transformers without Training

**Wuyang Chen**[1] , **Wei Huang**[2] , **Xianzhi Du**[3], **Xiaodan Song**[3], **Zhangyang Wang**[1], **Denny Zhou**[3]
[1]University of Texas, Austin    [2]University of Technology Sydney    [3]Google
{wuyang.chen,atlaswang}@utexas.edu    weihuang.uts@gmail.com
{xianzhi,xiaodansong,dennyzhou}@google.com

## Abstract

This work targets automated designing and scaling of Vision Transformers (ViTs). The motivation comes from two pain spots: 1) the lack of efficient and principled methods for designing and scaling ViTs; 2) the tremendous computational cost of training ViT that is much heavier than its convolution counterpart. To tackle these issues, we propose **As-ViT**, an auto-scaling framework for ViTs without training, which automatically discovers and scales up ViTs in an efficient and principled manner. Specifically, we first design a "seed" ViT topology by leveraging a training-free search process. This extremely fast search is fulfilled by a comprehensive study of ViT's network complexity, yielding a strong Kendall-tau correlation with ground-truth accuracies. Second, starting from the "seed" topology, we automate the scaling rule for ViTs by growing widths/depths to different ViT layers. This results in a series of architectures with different numbers of parameters in a single run. Finally, based on the observation that ViTs can tolerate coarse tokenization in early training stages, we propose a progressive tokenization strategy to train ViTs faster and cheaper. As a unified framework, **As-ViT** achieves strong performance on classification (83.5% top1 on ImageNet-1k) and detection (52.7% mAP on COCO) without any manual crafting nor scaling of ViT architectures: *the end-to-end model design and scaling process costs only 12 hours on one V100 GPU*. Our code is available at `https://github.com/VITA-Group/AsViT`.

## 1 Introduction

Transformer (Vaswani et al., 2017), a family of architectures based on the self-attention mechanism, is notable for modeling long-range dependencies in the data. The success of transformers has evolved from natural language processing to computer vision. Recently, Vision Transformer (ViT) (Dosovitskiy et al., 2020), a transformer architecture consisting of self-attention encoder blocks, has been proposed to achieve competitive performance to convolution neural networks (CNNs) (Simonyan & Zisserman, 2014; He et al., 2016) on ImageNet (Deng et al., 2009).

However, it remains elusive on how to effectively design, scale-up, and train ViTs, with three important gaps awaiting. First, Dosovitskiy et al. (2020) directly hard-split the 2D image into a series of local patches, and learn the representation with a pre-defined number of attention heads and channel expansion ratios. These ad-hoc "tokenization" and embedding mainly inherit from language tasks (Vaswani et al., 2017) but are not customized for vision, which calls for more flexible and principled **designs**. Second, the learning behaviors of ViT, including (loss of) feature diversity (Zhou et al., 2021), receptive fields (Raghu et al., 2021) and augmentations (Touvron et al., 2020; Jiang et al., 2021), differ vastly from CNNs. Benefiting from self-attention, ViT can capture global information even with shallow layers, yet its performance is quickly plateaued as going deeper. Strong augmentations are also vital to avoid ViTs from overfitting. These observations indicate that ViT architectures may require uniquely customized **scaling-up** laws to learn a more meaningful representation hierarchy. Third, training ViTs is both data and computation-consuming. To achieve state-of-the-art performance, ViT requires up to 300 million images and thousands of TPU-days. Although recent works attempt to enhance ViT's data and resource efficiency (Touvron et al., 2020; Hassani et al., 2021; Pan et al., 2021; Chen et al., 2021d), the heavy computation cost (e.g., quadratic with respect to the number of tokens) is still overwhelming, compared with training CNNs.

We point out that the above gaps are inherently connected by the core architecture problem: how to design and scale-up ViTs? Different from the convolutional layer that directly digests raw pixels, ViTs embed coarse-level local patches as input tokens. Shall we divide an image into non-overlapping tokens of smaller size, or larger but overlapped tokens? The former could embed more visual details in each token but ignores spatial coherency, while the latter sacrifices the local details but may benefit more spatial correlations among tokens. A further question is on ViT's depth/width trade-off: shall we prefer a wider and shallower ViT, or a narrower but deeper one? A similar dilemma also persists for ViT training: reducing the number of tokens would effectively speed up the ViT training, but meanwhile might sacrifice the training performance if sticking to coarse tokens from end to end.

In this work, we aim to reform the discovery of novel ViT architectures. Our framework, called **As-ViT** (Auto-scaling ViT), allows for extremely fast, efficient, and principled ViT design and scaling. In short, As-ViT first finds a promising "seed" topology for ViT of small depths and widths, then progressively "grow" it into different sizes (number of parameters) to meet different needs. Specifically, our "seed" ViT topology is discovered from a search space relaxed from recent manual ViT designs. To compare different topologies, we automate this process by a training-free architecture search approach and the measurement of ViT's complexity, which are extremely fast and efficient. This training-free search is supported by our comprehensive study of various network complexity metrics, where we find the expected length distortion has the best trade-off between time costs and Kendall-tau correlations. Our "seed" ViT topology is then progressively scaled up from a small network to a large one, generating a series of ViT variants in a single run. Each step, the increases of depth and width are automatically and efficiently balanced by comparing network complexities. Furthermore, to address the data-hungry and heavy computation costs of ViTs, we make our ViT tokens elastic, and propose a progressive re-tokenization method for efficient ViT training. We summarize our contributions as below:

1. We for the first time automate both the backbone design and scaling of ViTs. A "seed" ViT topology is first discovered (in only seven V100 GPU-hours), and then its depths and widths are grown with a principled scaling rule in a single run (five more V100 GPU-hours).

2. To estimate ViT's performance at initialization without any training, we conduct the first comprehensive study of ViT's network complexity measurements. We empirically find the expected length distortion has the best trade-off between the computation costs and its Kendall-tau correlations with ViT's ground-truth accuracy.

3. During training, we propose a progressive re-tokenization scheme via the change of dilation and stride, which demonstrates to be a highly efficient ViT training strategy that saves up to 56.2% training FLOPs and 41.1% training time, while preserving a competitive accuracy.

4. Our **As-ViT** achieves strong performance on classification (83.5% top-1 on ImageNet-1k) and detection (52.7% mAP on COCO).

## 2 WHY WE NEED AUTOMATED DESIGN AND SCALING PRINCIPLE FOR VIT?

**Background and recent development of ViT**[1]   To transform a 2D image into a sequence, ViT (Dosovitskiy et al., 2020) splits each image into $14 \times 14$ or $16 \times 16$ patches and embeds them into a fixed number of tokens; then following practice of the transformer for language modeling, ViT applies self-attention to learn reweighting masks as relationship modeling for tokens, and leverages FFN (Feed-Forward Network) layers to learn feature embeddings. To better facilitate the visual representation learning, recently works try to train deeper ViTs (Touvron et al., 2021; Zhou et al., 2021), incorporate convolutions (Wu et al., 2021; d'Ascoli et al., 2021; Yuan et al., 2021a), and design multi-scale feature extractions (Chen et al., 2021b; Zhang et al., 2021; Wang et al., 2021).

**Why manual design and scaling may be suboptimal?**   As the ViT architecture is still in its infant stage, there is no principle in its design and scaling. Early designs incorporate large token sizes, constant sequence length, and hidden size (Dosovitskiy et al., 2020; Touvron et al., 2020), and recent trends include small patches, spatial reduction, and channel doubling (Zhou et al., 2021; Liu et al., 2021). They all achieve comparably good performance, leaving the optimal choices unclear. Moreover, different learning behaviors of transformers from CNNs make the scaling law of ViTs

---

[1]We generally use the term "ViT" to indicate deep networks of self-attention blocks for vision problems. We always include a clear citation when we specifically discuss the ViTs proposed by Dosovitskiy et al. (2020).

highly unclear. Recent works (Zhou et al., 2021) demonstrated that attention maps of ViTs gradually become similar in deeper layers, leading to identical feature maps and saturated performance. ViT also generates more uniform representations across layers, enabling early aggregation of global context (Raghu et al., 2021). This is contradictory to CNNs as deeper layers help the learning of visual global information (Chen et al., 2018). These observations all indicate that previously studied scaling laws (depth/width allocations) for CNNs (Tan & Le, 2019) may not be appropriate to ViTs.

**What principle do we want?** We aim to automatically design and scale-up ViTs, being principled and avoiding manual efforts and potential biases. We also want to answer two questions: 1) Does ViT have any preference in its topology (patch sizes, expansion ratios, number of attention heads, etc.)? 2) Does ViT necessarily follow the same scaling rule of CNNs?

## 3 AUTO-DESIGN & SCALING OF VITS WITH NETWORK COMPLEXITY

To accelerate in ViT designing and avoid tedious manual efforts, we target efficient, automated, and principled search and scaling of ViTs. Specifically, we have two problems to solve: 1) with zero training cost (Section 3.2), how to efficiently find the optimal ViT architecture topology (Section 3.3)? 2) how to scale-up depths and widths of the ViT topology to meet different needs of model sizes (Section 3.4)?

### 3.1 EXPANDED TOPOLOGY SPACE FOR VITS

Before designing and scaling, we first briefly introduce our expanded topology search space for our As-ViT (*blue italics* in Figure 1). We first embed the input image into patches of a $\frac{1}{4}$-scale resolution, and adopt a stage-wise spatial reduction and channel doubling strategy. This is for the convenience of dense prediction tasks like detection that require multi-scale features. Table 1 summarizes details of our topology space, and will be explained below.

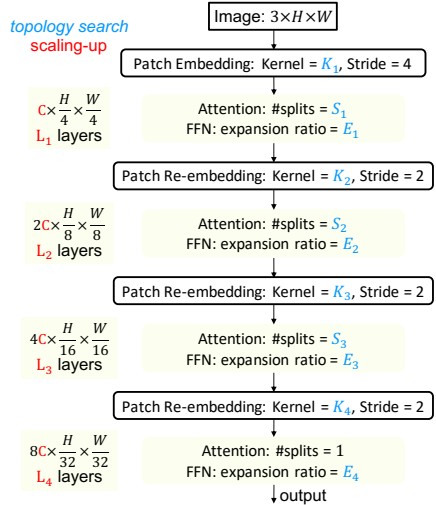

Figure 1: Overall architecture of our As-ViT. *Blue italics* indicates topology configurations to be searched (Table 1). Red indicates depth/width to be scaled-up.

**Elastic kernels.** Instead of generating non-overlapped image patches, we propose to search for the kernel size. This will enable patches to be overlapped with their neighbors, introducing more spatial correlations among tokens. Moreover, each time we downsample the spatial resolution, we also introduce overlaps when re-embedding local tokens (implemented by either a linear or a convolutional layer).

**Elastic attention splits.** Splitting the attention into local windows is an important design to reduce the computation cost of self-attention without sacrificing much performance (Zaheer et al., 2020; Liu et al., 2021). Instead of using a fixed number of splits, we propose to search for elastic attention splits for each stage[2]. Note that we try to make our design general and do not use shifted windows (Liu et al., 2021).

**More search dimensions.** ViT (Dosovitskiy et al., 2020) by default leveraged an FFN layer with $4\times$ expanded hidden dimension for each attention block. To enable a more flexible design of ViT architectures, for each stage we further search over the FFN expansion ratio. We also search for the final number of heads for the self-attention module.

Table 1: Topology Search Space for our As-ViT.

| Stage | Sub-space | Choices |
|---|---|---|
| #1 | Kernel $K_1$ | 4, 5, 6, 7, 8 |
| | Attention Splits $S_1$ | 2, 4, 8 |
| | FFN Expansion $E_1$ | 2, 3, 4, 5, 6 |
| #2 | Kernel $K_2$ | 2, 3, 4 |
| | Attention Splits $S_2$ | 1, 2, 4 |
| | FFN Expansion $E_2$ | 2, 3, 4, 5, 6 |
| #3 | Kernel $K_3$ | 2, 3, 4 |
| | Attention Splits $S_3$ | 1, 2 |
| | FFN Expansion $E_3$ | 2, 3, 4, 5, 6 |
| #4 | Kernel $K_4$ | 2, 3, 4 |
| | FFN Expansion $E_4$ | 2, 3, 4, 5, 6 |
| - | Num. Heads | 16, 32, 64 |

---

[2]Due to spatial reduction, the $4^{th}$ stage may already reach a resolution at $7 \times 7$ on ImageNet, and we set its splitting as 1.

### 3.2 ASSESSING VIT COMPLEXITY AT INITIALIZATION VIA MANIFOLD PROPAGATION

Training ViTs is slow: hence an architecture search guided by evaluating trained models' accuracies will be dauntingly expensive. We note a recent surge of training-free neural architecture search methods for ReLU-based CNNs, leveraging local linear maps (Mellor et al., 2020), gradient sensitivity (Abdelfattah et al., 2021), number of linear regions (Chen et al., 2021e;f), or network topology (Bhardwaj et al., 2021). However, ViTs are equipped with more complex non-linear functions: self-attention, softmax, and GeLU. Therefore, we need to measure their learning capacity in a more general way. In our work, we consider measuring the complexity of manifold propagation through ViT, to estimate how complex functions can be approximated by ViTs.

Intuitively, a complex network can propagate a simple input into a complex manifold at its output layer, thus likely to possess a strong learning capacity. In our work, we study the manifold complexity of mapping a simple circle input through the ViT: $\mathbf{h}(\theta) = \sqrt{N} \left[ \mathbf{u}^0 \cos(\theta) + \mathbf{u}^1 \sin(\theta) \right]$. Here, $N$ is the dimension of ViT's input (e.g. $N = 3 \times 224 \times 224$ for ImageNet images), $\mathbf{u}^0$ and $\mathbf{u}^1$ form an orthonormal basis for a 2-dimensional subspace of $\mathbb{R}^N$ in which the circle lives. We further define the ViT network as $\mathcal{N}$, its input-output Jacobian $\boldsymbol{v}(\theta) = \partial_\theta \mathcal{N}(\mathbf{h}(\theta))$ at the input $\theta$, and $\boldsymbol{a}(\theta) = \partial_\theta \boldsymbol{v}(\theta)$. We will calculate expected complexities over a certain number of $\theta$s uniformly sampled from $[0, 2\pi)$. In our work, we study three different types of manifold complexities:

**1. Curvature** can be defined as the reciprocal of the radius of the osculating circle on the ViT's output manifold. Intuitively, a larger curvature indicates that $\mathcal{N}(\theta)$ changes fast at a certain $\theta$. According to Riemannian geometry (Lee, 2006; Poole et al., 2016), the curvature can be explicitly calculated as $\kappa = \int (\boldsymbol{v}(\theta) \cdot \boldsymbol{v}(\theta))^{-3/2} \sqrt{(\boldsymbol{v}(\theta) \cdot \boldsymbol{v}(\theta))(\boldsymbol{a}(\theta) \cdot \boldsymbol{a}(\theta)) - (\boldsymbol{v}(\theta) \cdot \boldsymbol{a}(\theta))^2} d\theta$.

**2. Length Distortion** in Euclidean space is defined as $\mathcal{L}^E = \frac{\text{length}(\mathcal{N}(\theta))}{\text{length}(\theta)} = \int \sqrt{\|\boldsymbol{v}(\theta)\|_2} d\theta$. It measures when the network takes a unit-length curve as input, what is the length of the output curve. Since the ground-truth function we want to estimate (using $\mathcal{N}$) is usually very complex, one may also expect that networks with better performance should also generate longer outputs.

**3.** The problem of $\mathcal{L}^E$ is that, stretched outputs not necessarily translate to complex outputs. A simple example: even an appropriately initialized linear network could grow a straight line into a long output (i.e. a large norm of input-output Jacobian). Therefore, one could instead use **Length Distortion taking curvature into consideration** to measure how quickly the normalized Jacobian $\hat{\mathbf{v}}(\theta) = \mathbf{v}(\theta)/\sqrt{\mathbf{v}(\theta) \cdot \mathbf{v}(\theta)}$ changes with respect to $\theta$, defined as $\mathcal{L}^E_\kappa = \int \sqrt{\|\partial_\theta \hat{\mathbf{v}}(\theta)\|_2} d\theta$.

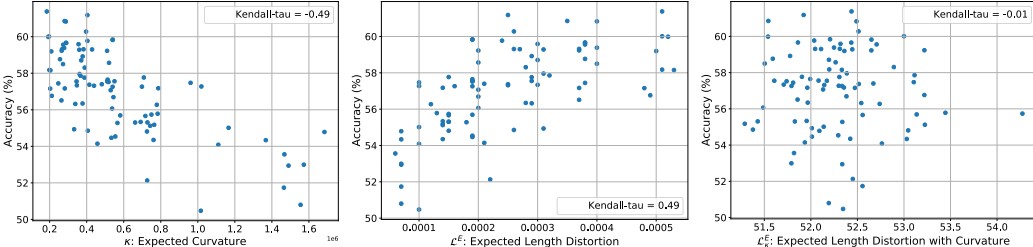

Figure 2: Correlations between $\kappa, \mathcal{L}^E, \mathcal{L}^E_\kappa$ and trained accuracies of ViT topologies from our search space.

In our study, we aim to compare the potential of using these three complexity metrics to guide the ViT architecture selection. As the core of neural architecture search is to rank the performance of different architectures, we measure the Kendall-tau correlations ($\tau$) between these metrics and models' ground-truth accuracies. We randomly sampled 87 ViT topologies from Table 1 (with $L_1 = L_2 = L_3 = L_4 = 1, C = 32$), fully train them on ImageNet-1k for 300 epochs (following the same training recipe of DeiT (Touvron et al., 2020)), and also measure their

Table 2: Complexity Study. $\tau$: Kendall-tau correlation. Time: per ViT topology on average on 1 V100 GPU.

| Complexity | $\tau$ | Time |
|---|---|---|
| $\kappa$ | -0.49 | 38.3s |
| $\mathcal{L}^E$ | 0.49 | 12.8s |
| $\mathcal{L}^E_\kappa$ | -0.01 | 48.2s |

$\kappa, \mathcal{L}^E, \mathcal{L}^E_\kappa$ at initialization. As shown in Figure 2, we can clearly see that both $\kappa$ and $\mathcal{L}^E$ exhibit high Kendall-tau correlations. $\kappa$ has a negative correlation, which may indicate that changes of output manifold on the tangent direction are more important to ViT training, instead of on the perpendicular direction. Meanwhile, $\kappa$ costs too much computation time due to second derivatives. We decide to choose $\mathcal{L}^E$ as our complexity measure for highly fast ViT topology search and scaling.

### 3.3 $\mathcal{L}^E$ AS REWARD FOR SEARCHING ViT TOPOLOGIES

We now propose our training-free search based on $\mathcal{L}^E$ (Algorithm 1). Most NAS (neural architecture search) methods evaluate the accuracies or loss values of single-path or super networks as proxy inference. This training-based search will suffer from more computation costs when applied to ViTs. Instead of training ViTs, for each architecture we sample, we calculate $\mathcal{L}^E$ and treat it as the reward to guide the search process. In addition to $\mathcal{L}^E$, we also include the NTK condition number $\kappa_\Theta = \frac{\lambda_{\max}}{\lambda_{\min}}$ to indicate the trainability of ViTs (Chen et al., 2021e; Xiao et al., 2019; Yang, 2020; Hron et al., 2020). $\lambda_{\max}$ and $\lambda_{\min}$ are the largest and smallest eigenvalue of NTK matrix $\Theta$.

---

**Algorithm 1:** Training-free ViT Topology Search.

---

1 **Input:** RL policy $\pi$, step $t = 0$, total steps $T$.
2 **while** $t < T$ **do**
3      Sample topology $\boldsymbol{a}_t$ from $\pi$.
4      Calculate $\mathcal{L}_t^E$ and $\kappa_{\Theta,t}$ for $\boldsymbol{a}_t$.
5      Normalization: $\widehat{\mathcal{L}}_t^E = \frac{\mathcal{L}_t^E - \mathcal{L}_{t-1}^E}{\max_{t'} \mathcal{L}_{t'}^E - \min_{t'} \mathcal{L}_{t'}^E}, \widehat{\kappa}_{\Theta,t} = \frac{\kappa_{\Theta,t} - \kappa_{\Theta,t-1}}{\max_{t'} \kappa_{\Theta,t'} - \min_{t'} \kappa_{\Theta,t'}}, t' = 1, \cdots, t.$
6      Update policy $\pi$ using reward $r_t = \widehat{\mathcal{L}}_t^E - \widehat{\kappa}_{\Theta,t}$ by policy gradient (Williams, 1992).
7      $t = t + 1$.
8 **return** *Topology $\boldsymbol{a}^*$ of highest probability from $\pi$.*

---

We use reinforcement learning (RL) for search. The RL policy is formulated as a joint categorical distribution over the choices in Table 1, and is updated by policy gradient (Williams, 1992). We update our policy for 500 steps, which is observed enough for the policy to converge (entropy drops from 15.3 to 5.7). The search process is extremely fast: only seven GPU-hours (V100) on ImageNet-1k, thanks to the fast calculation of $\mathcal{L}^E$ that bypasses the ViT training. To address the different magnitude of $\mathcal{L}^E$ and $\kappa_\Theta$, we normalize them by their relative value ranges (line 5 in Algorithm 1). We summarize the ViT topology statistics from our search in Table 3. We can see that $\mathcal{L}^E$ and $\kappa_\Theta$ highly prefer: (1) tokens with overlaps ($K_1 \sim K_4$ are all larger than strides), and (2) larger FFN expansion ratios in deeper layers ($E_1 < E_2 < E_3 < E_4$). No clear preference of $\mathcal{L}^E$ and $\kappa_\Theta$ are found on attention splits and number of heads.

Table 3: Statistics of topology search. *Standard deviation is normalized by mean due to different value ranges.

| Search Space | Mean | Std* |
|:---:|:---:|:---:|
| $K_1$ | 7.3 | 0.1 |
| $K_2$ | 4 | 0 |
| $K_3$ | 4 | 0 |
| $K_4$ | 4 | 0 |
| $E_1$ | 3.3 | 0.4 |
| $E_2$ | 3.9 | 0.4 |
| $E_3$ | 4.2 | 0.3 |
| $E_4$ | 5.2 | 0.2 |
| $S_1$ | 4 | 0.6 |
| $S_2$ | 2.7 | 0.5 |
| $S_3$ | 1.5 | 0.3 |
| Head | 42.7 | 0.5 |

### 3.4 AUTOMATIC AND PRINCIPLED SCALING OF ViTs

After obtaining an optimal topology, another question is: how to balance the network depth and width? Currently, there is no such rule of thumb for ViT scaling. Recent works try to scale-up or grow convolutional networks of different sizes to meet various resource constraints (Liu et al., 2019a; Tan & Le, 2019). However, to automatically find a principled scaling rule, training ViTs will cost enormous computation costs. It is also possible to search different ViT variants (as in Section 3.3), but that requires multiple runs. Instead, "scaling-up" is a more natural way to generate multiple model variants in one experiment. We are therefore motivated to scale-up our searched basic "seed" ViT to a larger model in an efficient training-free and principled manner.

We depict our auto-scaling method in Algorithm 2. The starting-point architecture has one attention block for each stage, and an initial hidden dimension $C = 32$. In each iteration, we greedily find the optimal depth and width to scale-up next. For depth, we try to find out which stage to deepen (i.e., add one attention block to which stage); for width, we try to discover the best expansion ratio (i.e., widen the channel number to what extent). The rule to choose how to scale-up is by comparing the propagation complexity among a set of scaling choices. For example, in the case of four backbone stages (Table 1) and four expansion ratio choices ($[0.05\times, 0.1\times, 0.15\times, 0.2\times]$), we have $4 \times 4 = 16$ scaling choices in total for each step. We calculate $\mathcal{L}^E$ and $\kappa_\Theta$ after applying each choice, and the one with the best $\mathcal{L}^E$ / $\kappa_\Theta$ trade-off (minimal sum of rankings by $\mathcal{L}^E$ and $\kappa_\Theta$) will be selected to scale-up with. The scaling stops when a certain limit of parameter number is reached. In our work, we stop the scaling process once the number of parameters reaches 100 million, and the scaling only takes five GPU hours (V100) on ImageNet-1k.

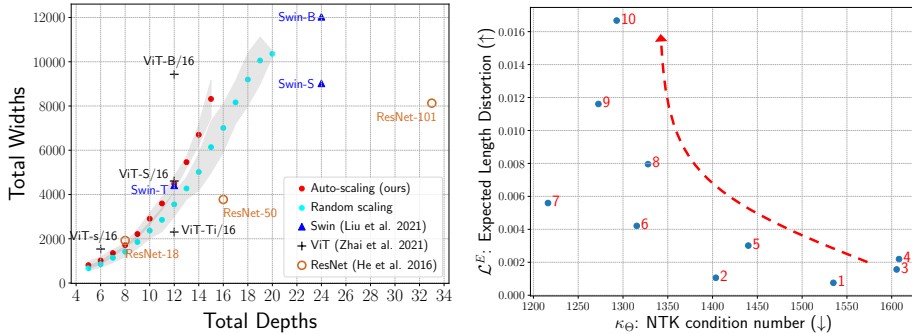

Figure 3: Left: Comparing scaling rules from As-ViT, random scaling, Swin (Liu et al., 2021), ViT (Zhai et al., 2021), and ResNet (He et al., 2016). "Total Depths": number of blocks ("bottleneck" of ResNet, "attention-block" of ViTs). "Total Widths": sum of output channel numbers from all blocks. Grey areas indicate standard deviations from 10 runs with different random seeds. Right: During the auto-scaling, both the network's complexity and trainability improve (numbers indicate scaling-up steps, $\mathcal{L}^E$ higher the better, $\kappa_\Theta$ lower the better).

The scaling trajectory is visualized in Figure 3. By comparing our automated scaling against random scaling, we find our scaling principle prefers to sacrifice the depths to win more widths, keeping a shallower but wider network. Our scaling is more similar to the rule developed by Zhai et al. (2021). In contrast, ResNet and Swin Transformer (Liu et al., 2021) choose to be narrower and deeper.

---

**Algorithm 2:** Training-free Auto-Scaling ViTs.

1 **Input:** seed As-ViT topology $\boldsymbol{a}_0$, stop criterion (#parameters) $P$, $t = 0$,
  channel expansion ratio choices $\mathcal{C} = \{1.05\times, 1.1\times, 1.15\times, 1.2\times\}$ (to increase the width by 5%, 10%, 15%, or 20%), depth choices $\mathcal{D} = \{(+1, 0, 0, 0), (0, +1, 0, 0), (0, 0, +1, 0), (0, 0, 0, +1)\}$ (to add one more layer to one of the four stages in Table 1).
2 **while** $P > $ *number of parameters of* $\boldsymbol{a}_t$ **do**
3      **for** *each scaling choice* $g_i \in \mathcal{C} \times \mathcal{D}$ **do**
4          Scale-up: $\boldsymbol{a}_{t,i} = \boldsymbol{a}_t \leftarrow g_i$.          ▷ Grow both the channel width and depth.
5          Calculate $\mathcal{L}_i^E$ and $\kappa_{\Theta,i}$ for $\boldsymbol{a}_{t,i}$.
6      Get ranking of each scaling choice $r_{\mathcal{L},i}$ by descendingly sort $\mathcal{L}_i^E$, $i = 1, \cdots, |\mathcal{C} \times \mathcal{D}|$.
7      Get ranking of each scaling choice $r_{\kappa_\Theta,i}$ by ascendingly sort $\kappa_{\Theta,i}$, $i = 1, \cdots, |\mathcal{C} \times \mathcal{D}|$.
8      Ascendingly sort each scaling choice $g_i$ by $r_{\mathcal{L}^E,i} + r_{\kappa_\Theta,i}$.
9      Select the scaling choice $g_i^*$ with the top (smallest) ranking.
10      $\boldsymbol{a}_{t+1} = \boldsymbol{a}_t \leftarrow g_i^*$.
11      $t = t + 1$.
12 **return** *Growed ViT architectures* $\boldsymbol{a}_1, \boldsymbol{a}_2, \cdots, \boldsymbol{a}_t$.

---

## 4 EFFICIENT VIT TRAINING VIA PROGRESSIVE ELASTIC RE-TOKENIZATION

Recent works (Jia et al., 2018; Zhou et al., 2019; Fu et al., 2020) show that one can use mixed or progressive precision to achieve an efficient training purpose. The rationale behind this strategy is that, there exist some "short-cuts" on the network's loss landscape that can be manually created to bypass perhaps less important gradient descent steps, especially during early training phases. As in ViT, both self-attention and FFN have quadratic computation costs to the number of tokens. It is therefore natural to ask: do we need full-resolution tokens during the whole training process?

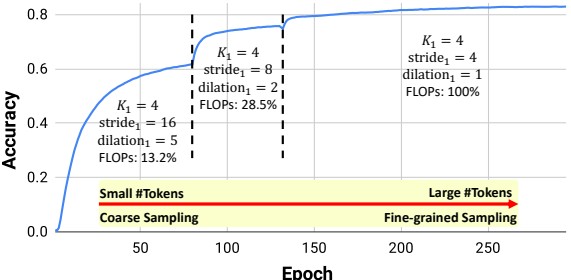

Figure 4: By progressively changing the sampling granularity (stride and dilation) of the first linear project layer, we can reduce the spatial resolutions of tokens and save training FLOPs (37.4% here), while still maintain a competitive final performance (ImageNet-1k $224 \times 224$). See Table 6 for more studies.

We provide an affirming answer by proposing a progressive elastic re-tokenization training strategy. To update the number of tokens during training without affecting the shape of weights in linear projections, we adopt different sampling granularities in the first linear projection layer. Taking the

first projection kernel $K_1 = 4$ with $\text{stride} = 4$ as an example: during training we gradually change the $(\text{stride}, \text{dilation})$ pair [3] of the first projection kernel to $(16, 5)$, $(8, 2)$, and $(4, 1)$, keeping the shape of weights and the architecture unchanged.

This re-tokenization strategy emulates curriculum learning for ViTs: when the training begins, we introduce coarse sampling to significantly reduce the number of tokens. In other words, our As-ViT quickly learns coarse information from images in early training stages at extremely low computation cost (only 13.2% FLOPs of full-resolution training). Towards the late phase of training, we progressively switch to fine-grained sampling, restore the full token resolution, and maintain the com-

Table 4: As-ViT topology and scaling rule.

| Design | Stage | K | S | E | Head |
|---|---|---|---|---|---|
| Seed Topology (*Blue italics* in Fig. 1) | #1 | 8 | 2 | 3 | 4 |
| | #2 | 4 | 1 | 2 | 8 |
| | #3 | 4 | 1 | 4 | 16 |
| | #4 | 4 | 1 | 6 | 32 |

| Scaling (Red in Fig. 1) | Stage-wise Depth | | | | Width ($C$) |
|---|---|---|---|---|---|
| | $L_1$ | $L_2$ | $L_3$ | $L_4$ | |
| As-ViT-Small | 3 | 1 | 4 | 2 | 88 |
| As-ViT-Base | 3 | 1 | 5 | 2 | 116 |
| As-ViT-Large | 5 | 2 | 5 | 2 | 180 |

petitive accuracy. As shown in Figure 4, when the ViT is trained with coarse sampling in early training phases, it can still obtain high accuracy while requiring extremely low computation cost. The transition between different sampling granularity introduces a jump in performance, and eventually the network restores its competitive final performance.

## 5 EXPERIMENTS

### 5.1 AS-VIT: AUTO-SCALING VIT

We show our searched As-ViT topology in Table 4. This architecture facilitates strong overlaps among tokens during both the first projection ("tokenization") step and three re-embedding steps. FFN expansion ratios are first narrow then become wider in deeper layers. A small number of attention splits are leveraged for better aggregation of global information.

The seed topology is automatically scaled-up, and three As-ViT variants of comparable sizes with previous works will be benchmarked. Our scaling rule prefers shallower and wider networks, and layers are more balanced among different resolution stages.

### 5.2 IMAGE CLASSIFICATION

**Settings.** We benchmark our As-ViT on ImageNet-1k (Deng et al., 2009). We use Tensorflow and Keras for training implementations and conduct all training on TPUs. We set the default image size as $224 \times 224$, and use AdamW (Loshchilov & Hutter, 2017) as the optimizer with cosine learning rate decay (Loshchilov & Hutter, 2016). A batch size of 1024, an initial learning rate of 0.001, and a weight decay of 0.05 are adopted.

Table 5 demonstrates comparisons of our As-ViT to other models. Compared to the previous both Transformer-based and CNN-based architectures, As-ViT achieves state-of-the-art performance with a comparable number of parameters and FLOPs.

Table 5: Image Classification on ImageNet-1k ($224 \times 224$).

| Method | Params. | FLOPs | Top-1 |
|---|---|---|---|
| RegNetY-4GF (Radosavovic et al., 2020) | 21.0 M | 4.0 B | 80.0% |
| ViT-S (Dosovitskiy et al., 2020) | 22.1 M | 9.2 B | 81.2% |
| DeiT-S (Touvron et al., 2020) | 22.0 M | 4.6 B | 79.8% |
| T2T-ViT-14 (Yuan et al., 2021b) | 21.5 M | 6.1 B | 81.7% |
| TNT-S (Han et al., 2021) | 23.8 M | 5.2 B | 81.5% |
| PVT-Small (Wang et al., 2021) | 24.5 M | 3.8 B | 79.8% |
| CaiT XS-24 (Touvron et al., 2021) | 26.6 M | 5.4 B | 81.8% |
| DeepVit-S (Zhou et al., 2021) | 27 M | 6.2 B | 82.3% |
| ConViT-S (d'Ascoli et al., 2021) | 27 M | 5.4 B | 81.3% |
| CvT-13 (Wu et al., 2021) | 20 M | 4.5 B | 81.6% |
| CvT-21 (Wu et al., 2021) | 32 M | 7.1 B | 82.5% |
| Swin-T (Liu et al., 2021) | 29.0 M | 4.5 B | 81.3% |
| BossNet-T0 (Li et al., 2021) | - | 3.4 B | 80.8% |
| AutoFormer-s (Chen et al., 2021c) | 22.9 M | 5.1 B | 81.7% |
| GLiT-Small (Chen et al., 2021a) | 24.6 M | 4.4 B | 80.5% |
| As-ViT Small (ours) | 29.0 M | 5.3 B | 81.2% |
| RegNetY-8GF (Radosavovic et al., 2020) | 39.0 M | 8.0 B | 81.7% |
| T2T-ViT-19 (Yuan et al., 2021b) | 39.2 M | 9.8 B | 82.2% |
| CaiT S-24 (Touvron et al., 2021) | 46.9 M | 9.4 B | 82.7% |
| ConViT-S+ (d'Ascoli et al., 2021) | 48 M | 10 B | 82.2% |
| ViT-S/16 (Dosovitskiy et al., 2020) | 48.6 M | 20.2 B | 78.1% |
| Swin-S (Liu et al., 2021) | 50.0 M | 8.7 B | 83.0% |
| DeepViT-L (Zhou et al., 2021) | 55 M | 12.5 B | 82.2% |
| PVT-Medium (Wang et al., 2021) | 44.2 M | 6.7 B | 81.2% |
| PVT-Large (Wang et al., 2021) | 61.4 M | 9.8 B | 81.7% |
| T2T-ViT-24 (Yuan et al., 2021b) | 64.1 M | 15.0 B | 82.6% |
| TNT-B (Han et al., 2021) | 65.6 M | 14.1 B | 82.8% |
| BossNet-T1 (Li et al., 2021) | - | 7.9 B | 82.2% |
| AutoFormer-b (Chen et al., 2021c) | 54 M | 11 B | 82.4% |
| ViT-ResNAS-t (Liao et al., 2021) | 41 M | 1.8 B | 80.8% |
| ViT-ResNAS-s (Liao et al., 2021) | 65 M | 2.8 B | 81.4% |
| As-ViT Base (ours) | 52.6 M | 8.9 B | 82.5% |
| RegNetY-16GF (Radosavovic et al., 2020) | 84.0 M | 16.0 B | 82.9% |
| ViT-B/16 (Dosovitskiy et al., 2020) [†] | 86.0 M | 55.4 B | 77.9% |
| DeiT-B (Touvron et al., 2020) | 86.0 M | 17.5 B | 81.8% |
| ConViT-B (d'Ascoli et al., 2021) | 86 M | 17 B | 82.4% |
| Swin-B (Liu et al., 2021) | 88.0 M | 15.4 B | 83.3% |
| GLiT-Base (Chen et al., 2021a) | 96.1 M | 17.0 B | 82.3% |
| ViT-ResNAS-m (Liao et al., 2021) | 97 M | 4.5 B | 82.4% |
| CaiT S-48 (Touvron et al., 2021) | 89.5 M | 18.6 B | 83.5% |
| As-ViT Large (ours) | 88.1 M | 22.6 B | 83.5% |

[†] Under $384 \times 384$ resolution.

---

[3] $\text{dilation} = round((\text{stride}/S_1 - 1) * K_1/(K_1 - 1)) + 1$, $S_1 = 4$ is the stride at the full token resolution.

More importantly, our As-ViT framework achieves competitive or stronger performance than concurrent NAS works for ViTs with much more search efficiency. As-ViTs are designed with highly reduced human or NAS efforts. All our three As-ViT variants are generated in only 12 GPU hours (on a single V100 GPU). In contrast, BoneNAS (Li et al., 2021) requires 10 GPU days to search a single architecture. For each variant of ViT-ResNAS (Liao et al., 2021), the super-network training takes 16.7∼21 hours, followed by another 5.5∼6 hours of evolutionary search.

**Efficient Training.** We leverage the progressive elastic re-tokenization strategy proposed in Section 4 to reduce both FLOPs and training time for large ViT models. As illustrated in Figure 4, we progressively apply $4\times$ and $2\times$ reductions on the number of tokens during training by changing both the dilation and the stride of the first linear projection layer. We tune the epochs allocated to each token

Table 6: Efficient training on ImageNet-1k ($224 \times 224$) via progressive elastic re-tokenization strategy (Section 4). $4\times$ (resp. $2\times$) indicates we reduce the number of tokens by 4 (resp. 2) times, and "N/A" indicates no token reduction.

| Token Reduction (Epochs) | | | FLOPs Saving | Training Time (TPU days) | Top1 Acc. |
|---|---|---|---|---|---|
| $4\times$ | $2\times$ | N/A | | | |
| 1∼40 | 41∼70 | 71∼300 | 18.7% | 36.9 | 83.1% |
| 1∼80 | 81∼140 | 141∼300 | 37.4% | 31.0 | 82.9% |
| 1∼120 | 121∼210 | 211∼300 | 56.2% | 25.2 | 82.5% |
| Baseline | | | 100% | 42.8 | 83.5 |

reduction stage and show the results in Table 6. Standard training takes 42.8 TPU days, whereas our efficient training could save up to 56.2% training FLOPs and 41.1% training TPU days, still achieving a strong accuracy.

**Disentangled Contributions from Topology and Scaling.** To better verify the contribution from our searched topology and scaling rule, we conduct more ablation studies (Table 7). First, we directly train the searched topology before scaling. Our searched seed topology is better than the best from 87 random topologies in Figure 2. Second, we compare our complexity-based scaling rule with "random scaling + As-ViT topology". At different scales, our automated scaling is also better than random scaling.

Table 7: Decoupling the contributions from the seed topology and the scaling, on ImageNet-1K.

| Model | Params. | FLOPs | Top-1 |
|---|---|---|---|
| As-ViT Topology | 2.4 M | 0.5 B | 61.7% |
| Random Topology | 2.2 M | 0.4 B | 61.4% |
| As-ViT Small | 29.0 M | 5.3 B | 81.2% |
| Random Scaling | 24.2 M | 8.7 B | 80.5% |
| As-ViT Base | 52.6 M | 8.9 B | 82.5% |
| Random Scaling | 42.4 M | 15.5 B | 82.2% |
| As-ViT Large | 88.1 M | 22.6 B | 83.5% |
| Random Scaling | 81.1 M | 28.7 B | 83.2% |

## 5.3 OBJECT DETECTION ON COCO

**Settings** Beyond image classification, we further evaluate our designed As-ViT on the detection task. Object detection is conducted on COCO 2017 that contains 118,000 training and 5000 validation images. We adopt the popular Cascade Mask R-CNN as the object detection framework for our As-ViT. We use an input size of $1024 \times 1024$, AdamW optimizer (initial learning rate of 0.001), weight decay of 0.0001, and a batch size of 256. Efficiently pretrained ImageNet-1K checkpoint (82.9% in Table 6) is leveraged as the initialization.

We compare our As-ViT to standard CNN (ResNet) and previous Transformer network (Swin (Liu et al., 2021)). The comparisons are conducted by changing only the backbones with other settings unchanged. In Table 8 we can see that our As-ViT can also capture multi-scale features and achieve state-of-the-art detection performance, although being designed on ImageNet and its complexity is measured for classification.

Table 8: Two-stage object detection and instance segmentation results. We compare employing different backbones with Cascade Mask R-CNN on single model without test-time augmentation.

| Backbone | Resolution | FLOPs | Params. | $AP_{val}$ | $AP_{val}^{mask}$ |
|---|---|---|---|---|---|
| ResNet-152 | 480∼800×1333 | 527.7 B | 96.7 M | 49.1 | 42.1 |
| Swin-B (Liu et al., 2021) | 480∼800×1333 | 982 B | 145 M | 51.9 | 45 |
| As-ViT Large (ours) | 1024×1024 | 1094.2 B | 138.8 M | 52.7 | 45.2 |

## 6 RELATED WORKS

### 6.1 VISION TRANSFORMER

Transformers (Vaswani et al., 2017) leverage the self-attention to extract global correlation, and become the dominant models for natural language processing (NLP) (Devlin et al., 2018; Radford et al., 2018; Brown et al., 2020; Liu et al., 2019b). Recent works explored transformers to vision problems: image classification (Dosovitskiy et al., 2020), object detection (Carion et al., 2020; Zhu et al., 2020; Zheng et al., 2020; Dai et al., 2020; Sun et al., 2020), segmentation (Chen et al., 2020; Wang et al., 2020), etc. The Vision Transformer (ViT) (Dosovitskiy et al., 2020) designed a pure transformer architecture and achieved SOTA performance on image classification. However, ViT heavily relies on large-scale datasets (ImageNet-21k (Deng et al., 2009), JFT-300M (Sun et al., 2017)) for pretraining, requiring huge computation resources. DeiT (Touvron et al., 2020) proposed Knowledge Distillation (KD) (Hinton et al., 2015; Yuan et al., 2020) via a special KD token to improve both performance and training efficiency. In contrast, our proposed As-ViT introduces more flexible tokenization, attention splitting, and FFN expansion strategies, with automated discovery.

### 6.2 NEURAL ARCHITECTURE DESIGN AND SCALE

Manual design of network architectures heavily relies on human prior, which is difficult to scale-up. Recent works leverage AutoML to find optimal combinations of operators/topology in a given search space (Zoph & Le, 2016; Real et al., 2019; Liu et al., 2018; Dong & Yang, 2019). However, the searched model are small due to the fixed and hand-crafted search space, far from being scaled-up to modern networks. For example, models from NASNet space (Zoph et al., 2018) only have ∼5M parameters, much smaller than real-world ones (20 to over 100M). One main reason for not being scalable is because NAS is a computation-consuming task, typically costing 1∼2 GPU days to search even small architectures. Meanwhile, many works try to grow a "seed" architecture to different variants. EfficientNet (Tan & Le, 2019) manually designed a scaling rule for width and depth. Give a template backbone with fixed depth, Liu et al. (2019a) grow the width by gradient descent. For ViT, we for the first time bring both architecture design and scaling together in one framework. To overcome the computation-consuming problem in the training of transformers, we directly use the complexity of manifold propagation as a surrogate measure towards a training-free search and scale.

### 6.3 EFFICIENT TRAINING

A number of methods have been developed to accelerate the training of deep neural networks, including mixed precision (Jia et al., 2018), distributed optimization (Cho et al., 2017), large-batch training (Goyal et al., 2017; Akiba et al., 2017; You et al., 2018), etc. Jia et al. (2018) combined distributed training with a mixed-precision framework. Wang et al. (2019) proposed to save deep CNN training energy cost via stochastic mini-batch dropping and selective layer update. In our work, customized progressive tokenization via the changing of stride/dilation can effectively reduce the number of tokens during ViT training, thus largely saving the training cost.

## 7 CONCLUSIONS

To automate the principled design of vision transformers without tedious human efforts, we propose As-ViT, a unified framework that searches and scales ViTs without any training. Compared with hand-crafted ViT architecture, our As-ViT leverages more token overlaps, increased FFN expansion ratios, and is wider and shallower. Our As-ViT achieves state-of-the-art accuracies on both ImageNet-1K classification and COCO detection, which verifies the strong performance of our framework. Moreover, with progressive tokenization, we can train heavy ViT models with largely reduced training FLOPs and time. We hope our methodology could encourage the efficient design and training of ViTs for both the transformer and the NAS communities.

### ACKNOWLEDGEMENT

Z.W. is in part supported by the NSF AI Institute for Foundations of Machine Learning (IFML) and a Google TensorFlow Model Garden Award.

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

## A  IMPLEMENTATIONS

**Training-free Topology Search and Scaling.**  We calculate $\mathcal{L}^E$ by uniformly sampling 10 $\theta$s in $[0, 2\pi)$. For one architecture, the calculation of $\mathcal{L}^E$ is repeated five times with different (random) network initializations, and $\mathcal{L}^E$ is set as their mean.

**Image Classification.**  We use 20 epochs of linear warm-up, a batch size of 1,024, an initial learning rate of 0.001, and a weight decay of 0.05. Augmentations including stochastic depth (Huang et al., 2016), Mixup (Zhang et al., 2017), Cutmix (Yun et al., 2019), RandAug (Cubuk et al., 2020), Exponential Moving Average (EMA) are also applied.

**Object Detection.**  Our training adopts a batch size of 256 for 36 epochs, with also stochastic depth. We do not use any stronger techniques like HTC (Chen et al., 2019), multi-scale testing, sotf-NMS (Bodla et al., 2017), etc.

## B  CONVERGENCE OF TRAINING-FREE SEARCH

To demonstrate the convergence of the policy learned by our RL search, we show the entropy during learning the policy in Figure 5. We can see that a training of 500 steps is enough for the policy to converge to low entropy (high confidence).

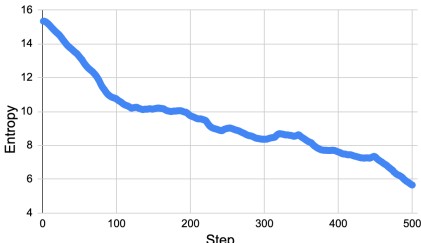

Figure 5: Entropy of policy during our search (Section 3.3).

