# OpenReview forum: "Auto-scaling Vision Transformers without Training"
_ICLR.cc/2022/Conference — ICLR 2022 Poster_

### Official Review · Reviewer_rFuj · 2021-11-01

**Correctness:** 4
**Technical Novelty And Significance:** 3
**Empirical Novelty And Significance:** 4
**Recommendation:** 8
**Confidence:** 4

**Main Review:**

Pros:
-1- As-ViT is a novel all-in-one efficient framework: it first finds a promising “seed” topology for ViT of minimal depths and widths, then progressively “grow” it into different s capacities to meet different needs, and finally “train” the desired architecture. All three steps are coherently designed to be very efficient, which is important for designing and verifying ViTs.
-2- For searching ViTs, the seed topology space is relaxed from recent vanilla ViT designs. To compare different topologies, the authors automate this process by a training-free architecture search approach via measuring the network complexity, which is extremely fast and efficient.
While several previous works already explored NAS for CNN with gradient features or linear regions, transformer has more nonlinear structures and different components. The authors introduced a new general complexity metric of manifold propagation that can be applicable to transformers. The proposed training-free search is experimentally supported by a comprehensive study involving measure time costs and Kendall-tau correlations.
-3- The “seed” ViT topology is then progressively scaled up from a small to a large network, generating a series of ViT variants in a single run. At each step, how to increase the depth and width is automatically determined by comparing network complexities of different scaling choices, being both principled and efficient. The authors claimed they discovered ViT scaling principles that echo the empirical observation of (Zhai 2021).
-4- Further, to address the heavy training costs of ViTs, the authors proposed to make ViT tokens elastic, an idea similar to dynamically downsampling image resolution for efficient CNN training. They then propose a progressive re-tokenization method for efficient ViT training. The idea is not particularly novel but is the first time of this type in ViTs.
-5- The authors made comprehensive comparisons with various SOTA ViTs and their NAS methods, on not only ImageNet-1K classification but also COCO detection. As-ViT performs better than or competitively to most of them, meanwhile largely saving training FLOPs and time so that is also impressive. However, I have some reservations regarding their discussion of results, as to be laid out below.

The paper writing is overall good. The results and findings suggest potentially broad impact of this work.

Cons:
-1- Comparison to Prior Arts of NAS for ViTs
My main issue is that, from table 5, I cannot conclude the same as the authors stated “As-ViT framework significantly outperforms concurrent NAS works for ViTs in both performances and search efficiency”. For example, BossNet has marginal inferior accuracies but also smaller models; the more recent AutoFormer has also the same accuracies for Small/Base, only costing more FLOPs at Base level.
I understand that their search space comparison is not apple-to-apple (e.g.,  BossNet used conv layers too); and As-ViT has other selling points such as efficient search and training. However, the authors are urged to put their accuracy comparison in the fairer and more detailed context, and the current conclusions look a bit over-rush and misleading to me.
-2- Comparison to Swin Transformers
For image classification, the reported performance of As-Vit is slight less competitive than Swin at comparable costs. Again, I understand the two are not fully apple-to-apple comparable due to vastly different building blocks, and I am not asking As-ViT to outperform Swin since the former’s current search space is more vanilla.  Yet, as Swin follows a different scaling philosophy to be narrower and deeper, how can you then be convinced that your discovered “shallower but wider” principle is optimal?
Also, in object detection, As-ViT Large seems to outperform Swin-B, but the comparison is questionable to me: why the image resolutions are not aligned between the two? If Swin on averages sees smaller image inputs, it can possibly explain why its FLOPs and AP are both lower than As-ViT, and a fairer comparison should be appended.
Further, the current As-ViT space searches for attention splits, which seems to be a simplified sliding window without overlapping. Then, could you expand your search space to include Swin-type architecture into your search space? To be fair, I am not asking the authors to perform so during the rebuttal timeframe. I’m instead asking to address the above fairness issues.


**Summary Of The Paper:**

The paper proposed As-ViT to automate the principled design of vision transformers without tedious human efforts. To my best knowledge, it is the first framework that unifies efficient search, scaling and training in ViTs. The empirical performance is in general satisfactory.

**Summary Of The Review:**

The biggest merit of this paper is to unify efficient search, scaling and training as one -- that is unique and strong. The results prove their concepts and can be considered as promising since the adopted search space is vanilla. However, the authors are expected to clarify a number of experiment and comparison issues as aforementioned.

---

> ### Author Response · Authors · 2021-11-15
> **Response to reviewer rFuj's questions**
>
> We sincerely appreciate the rFuj’s review.
>
> **“Cons: -1-”** We greatly appreciate this suggestion, and we have tune-downed our claim in the updated submission, as shown in blue lines in the revised PDF.
>
> **“Cons: -2-”** We would like to clarify that we are not claiming our scaling rule is universally optimal. Instead, our scaling rule shows strong performance on the topology we discovered. This is verified by studying the decoupled contributions from our scaling in Table 7. We have revised our submission to make our claim more clear.
> * For the input size, the area of maximal input size in our work (1024x1024=1048576) is comparable to that in Swin (800*1280=1024000, see https://github.com/SwinTransformer/Swin-Transformer-Object-Detection/blob/master/tools/analysis_tools/get_flops.py#L21). This is indeed the reason we carefully choose the 1024x1024 input size to make the comparison fair. Also, we would like to clarify that the choice of input size is just different training conventions on COCO detection. There are many detection works that use square inputs.
> * Our experiments are based on the Google Cloud TPU detection codebase. This codebase by default sets square input sizes, since TensorFlow builds static computation graphs and does not allow changing the input size (part of the graph) once the model is built. Please see an example configuration here (from the published SpineNet [CVPR’20] also implemented in TensorFlow):
> https://github.com/tensorflow/tpu/blob/master/models/official/detection/configs/spinenet/spinenet96_mrcnn.yaml#L26
> * For the search space: we appreciate this suggestion. The reason we did not include the shifted window in Swin is to make our search space simple and neat, without introducing highly customized operators. We will also try to expand our search space to include the shifted window.

---

### Official Review · Reviewer_eYy9 · 2021-11-02

**Correctness:** 3
**Technical Novelty And Significance:** 3
**Empirical Novelty And Significance:** 3
**Recommendation:** 6
**Confidence:** 4

**Details Of Ethics Concerns:**

No Ethics concerns to this paper.

**Main Review:**

Strengths:

+The problem is significant

ViT training is time-consuming and the input resolution is fixed for a specific ViT model, which makes it important to have a proper scaling rule for choosing the size of ViT. The search-based auto-scaling rule proposed in this paper is a good way to solve this issue.

+NAS on ViT

The author uses neural architecture search on ViT-like framework on the number of kernels, attention splits, expansion ratio, depth and width jointly. The searched architecture are stage-wise and can be used for down-stream tasks like detection, which is another technical contribution of this paper.

Weaknesses:

-Comparison with other methods

Compared with original single-scale ViT or a multi-scale vision transformer like Swin, the proposed AS-ViT does not outperform them (ViT-S 81.2%, Swin-T 81.3%, AS-S 81.2%; Swin-S 83.0%, AS-B 82.5%; Swin-B 83.3%, AS-L 83.5%). Swin here can be viewed as an augmented multi-scale vision transformer. The author probably needs to propose a Multi-scale ViT baseline for comparison here and show the effectiveness of the searched AS-ViT.

**Summary Of The Paper:**

In this paper, the author proposes auto-scaling ViT, which is to search seed ViT topology based on number of kernels, attention splits, expansion ratio, depth and width jointly. The searched ViT is trained with a progressive re-tokenization scheme that saves ~40% training time and preserves the accuracy with less parameters. Experiments on ImageNet and MSCOCO show that AS-ViT can reach strong performances on classification and detection.

**Summary Of The Review:**

The paper AS-ViT is focusing on solving the scaling issue of ViT with searched topology and auto-scaling rules. However, it does not show significant improvement compared with other multi-scale vision transformers and my concern is that a multi-scale ViT baseline for fair comparison and evaluation is needed here. I will increase my rating if the concerns are well addressed here.

---

> ### Author Response · Authors · 2021-11-15
> **Response to reviewer eYy9's questions**
>
> We sincerely thank the eYy9’s review.
>
> We first would like to clarify a possible misunderstanding: our AsViT is a multi-scale ViT model. AsViT has four stages, with each extracting features of a specific resolution. These multi-scale pyramid features are fed into the detection head (FPN in our work), and they are vital to our high detection performance (52.7% mAP vs. Swin 51.9%).
>
> Meanwhile, we respectfully and kindly suggest if eYy9 could spend more time carefully reading our work, since our contribution is much more beyond a single table of accuracies, see point 2~4 below:
> 1. Our AsViT shows strong performance. On ImageNet, AsViT outperforms many ViT works published in ICCV’21: GLiT-Small 80.5%, PVT-Small 79.8%, vs. AS-S 81.2%; AutoFormer-b 82.4%, PVT-Large 81.7%, vs. AS-B 82.5%; and GLiT-Base 82.3%, ViT-ResNAS-m 82.4%. Without using shifted window and relative position bias, our larger mode (83.5%) can outperform Swin Transformer (83.3%), and also on the COCO detection task (52.7% mAP vs. Swin 51.9%).
> We are also in the process of trying other suggested models and will update once the results are ready.
> 2. We provide the first comprehensive study on measuring ViT’s network complexity, showing feasible estimations of ViT’s performance at initialization with strong correlations against the accuracies after training. This contribution is confirmed by reviewer rFuj (“Pros -2-”).
> Understanding how the architectural properties (depth, width, etc.) affect the neural network’s universal approximation and its ensuing performance is a foundational question, and there is a rich history of prior work addressing expressivity in neural networks [1, 2, 3, 4]. Our results demonstrate that general geometric analysis methods (curvature, length distortion) can indeed be applied to study ViT’s expressivity and thus can be useful tools to study structural properties of the ViT family.
> 3. Our complexity measurements lead to training-free ViT architecture search, which is an increasingly important NAS direction [5, 6, 7, 8, 9]. This contribution is also confirmed by reviewer rFuj (“Pros -3-”). Our framework makes breakthroughs in two directions: 1) we work on more challenging non-linear layers (GeLU and self-attention); 2) we not only search the basic topology but also scale it up to larger models.
> 4. As reviewer rFuj pointed (“Pros -1-”), our AsViT is a “novel all-in-one efficient framework”: we design and scale-up a series of ViT variants in a single job, with only 10 GPU hours in total.
>
> [1] Hornik, Kurt, Maxwell Stinchcombe, and Halbert White. "Multilayer feedforward networks are universal approximators." Neural networks 2, no. 5 (1989): 359-366.
>
> [2] Cybenko, George. "Approximation by superpositions of a sigmoidal function." Mathematics of control, signals and systems 2, no. 4 (1989): 303-314.
>
> [3] Bartlett, Peter L., Vitaly Maiorov, and Ron Meir. "Almost linear VC-dimension bounds for piecewise polynomial networks." Neural computation 10, no. 8 (1998): 2159-2173.
>
> [4] Pascanu, Razvan, Guido Montufar, and Yoshua Bengio. "On the number of response regions of deep feed forward networks with piece-wise linear activations." arXiv preprint arXiv:1312.6098 (2013).
>
> [5] Park, Daniel S., Jaehoon Lee, Daiyi Peng, Yuan Cao, and Jascha Sohl-Dickstein. "Towards nngp-guided neural architecture search." arXiv preprint arXiv:2011.06006 (2020).
>
> [6] Mellor, Joe, Jack Turner, Amos Storkey, and Elliot J. Crowley. "Neural architecture search without training." In International Conference on Machine Learning, pp. 7588-7598. PMLR, 2021.
>
> [7] Abdelfattah, Mohamed S., Abhinav Mehrotra, Łukasz Dudziak, and Nicholas D. Lane. "Zero-cost proxies for lightweight nas." arXiv preprint arXiv:2101.08134 (2021).
>
> [8] Chen, Wuyang, Xinyu Gong, and Zhangyang Wang. "Neural architecture search on imagenet in four gpu hours: A theoretically inspired perspective." arXiv preprint arXiv:2102.11535 (2021).
>
> [9] Xiang, Lichuan, Łukasz Dudziak, Mohamed S. Abdelfattah, Thomas Chau, Nicholas D. Lane, and Hongkai Wen. "Zero-Cost Proxies Meet Differentiable Architecture Search." arXiv preprint arXiv:2106.06799 (2021).

---

### Official Review · Reviewer_Bp2F · 2021-11-02

**Correctness:** 4
**Technical Novelty And Significance:** 3
**Empirical Novelty And Significance:** 2
**Recommendation:** 5
**Confidence:** 4

**Main Review:**



- A two-stage approach is followed for architecture search. First, there is a search for seed architecture using a training-free strategy. The seed ViT is found from existing designs. Second, the seed is scaled upon along width and depth.

- the paper also proposes elastic-tokens: progressive re-tokenization for fast ViT training.

- While the design search is efficient: requires only 7+5 GPU hours for seed+scaling stages. Compared with existing NAS based approaches, the proposed method requires less time to search for the optimal architecture. However, the final searched architecture by the proposed method has more FLOPs than existing methods e.g., ViT-ResNAS-t with 1.8B FLOPs compared with 8.9B of the proposed architecture, with slightly better performance.

**Summary Of The Paper:**

Initial variants of ViTs borrow their architecture from the seminal ‘attention is all you need’ work. Some recent methods amend the default transformer architecture to incorporate convolutions in the design (e.g., CCT: Compact Convolutional Transformers), or a hierarchical architecture with feature pyramids (e.g., Swin, PyramidViT etc) to make ViTs suitable for dense prediction tasks (detection and segmentation).
Instead of relying on hand-designed architectures, the paper proposes to automatically search for a ViT architecture, that is both efficient & accurate.

**Summary Of The Review:**


- It is unclear if the proposed strategy works to search a ViT architecture suitable for small-scale datasets?

- The advantages of automatically searched ViT architectures are not immediately clear. Their major limitation for ViTs is the time they require for training, and the number of FLOPs introduced by quadratic complexity in self-attention. Compared with existing “hand-designed” ViT architectures, the proposed automatically searched ViT architecture does not significantly reduce compute training time.

- I believe an automatically searched ViT architecture, that is efficient to train, will help democratize ViTs research, and make it accessible to broader resource-constrained academic labs.

---

> ### Author Response · Authors · 2021-11-15
> **Response to reviewer Bp2F's questions**
>
> We thank Bp2F’s review, but we respectfully disagree with your comments, as explained below:
> 1. Searching for a compact and efficient model largely relies on including efficient operations into the search space and budget-aware penalties on parameters/FLOPs [1, 2], which is beyond the scope of our work.
> 2. As confirmed by reviewer rFuj (“Pros -5-”), our AsViT outperforms many published ViT works in terms of efficiency, and many published ViT architectures designed by NAS or humans are also not compact, e.g.: “GLiT-Base [ICCV’21] 96.1M params. 82.3% accuracy” vs. “As-ViT Large 88.1M params. 83.5% accuracy”; “AutoFormer-b [ICCV’21] 54M params. 82.4% accuracy” vs. “As-ViT Base 52.6M params. 82.5% accuracy”.
>   - In addition, for the question “ViT-ResNAS-t with 1.8B FLOPs”, this is because they mainly focused on image classification and used 14x14 coarse input patches; in contrast, we use 4x4 fine-grained patches to target both image classification and object detection, and our 82.5% - 80.8% = 1.7% performance gain is not “slightly better” but a large gap. Meanwhile, if we also switch to 14x14 patches, our FLOPs is 1.41G, even smaller than ViT-ResNAS-t.
> 3. We respectfully disagree that having “quadratic complexity” makes the research topic “automatic designing and scaling of ViT architectures” less important. In fact, many published ViT-NAS works also leveraged vanilla self-attention layers (e.g. Fig. 3 in AutoFormer [ICCV’21], Fig. 3 in BossNAS [ICCV’21]), and achieved comparable or even higher computation costs than ours.
>   - More importantly, our work is versatile and contributes to the research community in multiple folds:
>
>     (a) To understand how the architectures affect the ViT’s **universal approximation**, we provide the first comprehensive study on measuring ViT’s complexity, showing feasible estimations of ViT’s performance at initialization with strong correlations against the accuracies after training (confirmed by reviewer rFuj (“Pros -2-”));
>
>     (b) Our complexity measurements lead to **training-free ViT architecture search**, which is an increasingly important NAS direction [3, 4, 5, 6, 7] (confirmed by reviewer rFuj (“Pros -3-”));
>
>     (c) As reviewer rFuj pointed (“Pros -1-”), our AsViT is a “novel all-in-one efficient framework”: we design and scale-up a series of ViT variants **in a single job**, with only 10 GPU hours in total.
> 4. We strongly believe there is an unfair comment: “Compared with existing “hand-designed” ViT architectures, the proposed automatically searched ViT architecture does not significantly reduce compute training time.” We **can** largely reduce 41.1% training time by using elastic tokenization with an 82.5% accuracy, which is still higher than some previous works (ViT-ResNAS-m 82.4%, GLiT-Base 82.3%, DeiT-B 81.8%).
> 5. Small-scale dataset: following the same training settings in DeiT, we also train our discovered As-VIT Large architecture on CIFAR-10, and our performance is 97.7% > DeiT-B 97.5%.
>
> [1] Wu, Bichen, Xiaoliang Dai, Peizhao Zhang, Yanghan Wang, Fei Sun, Yiming Wu, Yuandong Tian, Peter Vajda, Yangqing Jia, and Kurt Keutzer. "Fbnet: Hardware-aware efficient ConvNet design via differentiable neural architecture search." In Proceedings of the IEEE/CVF Conference on Computer Vision and Pattern Recognition, pp. 10734-10742. 2019.
>
> [2] Tan, Mingxing, Bo Chen, Ruoming Pang, Vijay Vasudevan, Mark Sandler, Andrew Howard, and Quoc V. Le. "Mnasnet: Platform-aware neural architecture search for mobile." In Proceedings of the IEEE/CVF Conference on Computer Vision and Pattern Recognition, pp. 2820-2828. 2019.
>
> [3] Park, Daniel S., Jaehoon Lee, Daiyi Peng, Yuan Cao, and Jascha Sohl-Dickstein. "Towards nngp-guided neural architecture search." arXiv preprint arXiv:2011.06006 (2020).
>
> [4] Mellor, Joe, Jack Turner, Amos Storkey, and Elliot J. Crowley. "Neural architecture search without training." In International Conference on Machine Learning, pp. 7588-7598. PMLR, 2021.
>
> [5] Abdelfattah, Mohamed S., Abhinav Mehrotra, Łukasz Dudziak, and Nicholas D. Lane. "Zero-cost proxies for lightweight nas." arXiv preprint arXiv:2101.08134 (2021).
>
> [6] Chen, Wuyang, Xinyu Gong, and Zhangyang Wang. "Neural architecture search on imagenet in four gpu hours: A theoretically inspired perspective." arXiv preprint arXiv:2102.11535 (2021).
>
> [7] Xiang, Lichuan, Łukasz Dudziak, Mohamed S. Abdelfattah, Thomas Chau, Nicholas D. Lane, and Hongkai Wen. "Zero-Cost Proxies Meet Differentiable Architecture Search." arXiv preprint arXiv:2106.06799 (2021).

---

> ### Author Response · Authors · 2021-11-23
> **Look forward to more discussions.**
>
> Dear reviewer Bp2F:
>
> We highly appreciate your suggestions and review efforts!
>
> In our response (below), we tried to address your concerns on performance, FLOPs, training time savings, and small-scale datasets. We also re-explained our contributions.
>
> We would appreciate it if you could please take a look and finalize your review on our work, hopefully more positively. We truly thank you again for the valued efforts, and look forward to more discussions!
>
> Sincerely,
> Paper2939 Authors

---

> ### Author Response · Authors · 2021-11-27
> **Look forward to further addressing your concerns.**
>
> Dear reviewer Bp2F:
>
> We very much look forward to further addressing your concerns at our best, and we highly appreciate your suggestions and review efforts!
>
> In our response below, we responded to your concerns on performance, FLOPs, training time savings, and small-scale datasets. We also tried to re-explained our contributions in a more clear way.
>
> We would highly appreciate it if you could please read our response and finalize your review on our work, hopefully more positively. Your review and suggestions are very important to us!
>
> Sincerely,
>
> Paper2939 Authors

---

### Author Response · Authors · 2021-11-29
**Summary of updates from Authors**

Dear Reviewers and AC panel,

Thank you again for your valuable reviews that have helped improve and revise our submission. We are happy that the contributions of our work have been recognized by reviewers eYy9 and rFuj. We also realize that the end of the rebuttal period is only one day away, and we are yet to hear from reviewer Bp2F as we tried to address the concerns with our response (which we believe can help better understand the merits of our work).

We would like to summarize the key points of our work and responses for a fast understanding:

(a) The **first** comprehensive study of **ViT’s model complexity**, showing feasible estimations of ViT’s performance at initialization with strong correlations with the accuracies after training (confirmed by reviewer rFuj).

(b) **Training-free ViT architecture design**, which is an increasingly important NAS direction [1, 2, 3, 4, 5] (confirmed by reviewer eYy9 and rFuj).

(c) **“All-in-one efficient framework”**: we design and scale up a series of ViT variants in a single job, with only 10 GPU hours in total (confirmed by reviewer eYy9 and rFuj).

(d) **Strong performance against published works**: ImageNet GLiT-Small 80.5%, PVT-Small 79.8%, vs. As-ViT-S 81.2%; AutoFormer-b 82.4%, PVT-Large 81.7%, vs. As-ViT-B 82.5%; Swin-B 83.3% vs. As-ViT-L 83.5%; plus on COCO detection task Swin 51.9% vs. As-ViT 52.7% mAP.

We sincerely hope in keeping a positive discussion on the contributions of our work. Please do not hesitate to contact us if there are any more concerns that we could address. Thanks!

Sincerely,

Authors

**References**

[1] Park, Daniel S., Jaehoon Lee, Daiyi Peng, Yuan Cao, and Jascha Sohl-Dickstein. "Towards nngp-guided neural architecture search." arXiv preprint arXiv:2011.06006 (2020).

[2] Mellor, Joe, Jack Turner, Amos Storkey, and Elliot J. Crowley. "Neural architecture search without training." In International Conference on Machine Learning, pp. 7588-7598. PMLR, 2021.

[3] Abdelfattah, Mohamed S., Abhinav Mehrotra, Łukasz Dudziak, and Nicholas D. Lane. "Zero-cost proxies for lightweight nas." arXiv preprint arXiv:2101.08134 (2021).

[4] Chen, Wuyang, Xinyu Gong, and Zhangyang Wang. "Neural architecture search on imagenet in four gpu hours: A theoretically inspired perspective." arXiv preprint arXiv:2102.11535 (2021).

[5] Xiang, Lichuan, Łukasz Dudziak, Mohamed S. Abdelfattah, Thomas Chau, Nicholas D. Lane, and Hongkai Wen. "Zero-Cost Proxies Meet Differentiable Architecture Search." arXiv preprint arXiv:2106.06799 (2021).

---

### Decision · Program_Chairs · 2022-01-20

**Decision:**

Accept (Poster)

**Comment:**

The paper introduces As-ViT, an interesting framework for searching and scaling ViTs without training. Overall, the paper received positive reviews. On the other hand, R1 rated the paper as marginally below the threshold, raising concerns about search on small datasets and issues regarding the comparison in terms of FLOPS/accuracy with other methods. The authors adequately addressed these concerns in the rebuttal, and helped clarify other questions by R2 and R3. R1 did not participate in the discussion after the author response nor updated his/her review. The AC agrees with R2 and R3 that the paper passes the acceptance bar of ICLR, as the unified approach for efficient search/scaling/training is novel and should be interesting to the ICLR audience.